# A Real-Time Channel Prediction Model Based on Neural Networks for Dedicated Short-Range Communications

**DOI:** 10.3390/s19163541

**Published:** 2019-08-13

**Authors:** Tianhong Zhang, Sheng Liu, Weidong Xiang, Limei Xu, Kaiyu Qin, Xiao Yan

**Affiliations:** 1School of Aeronautics and Astronautics, University of Electronic Science and Technology of China, Chengdu 611731, China; 2Department of Electrical and Computer Engineering, University of Michigan, Dearborn, MI 48128, USA

**Keywords:** channel models, neural networks, wireless communications, prediction methods, vehicular and wireless technologies, dedicated short-range communication

## Abstract

Based on a multiple layer perceptron neural networks, this paper presents a real-time channel prediction model, which could predict channel parameters such as path loss (PL) and packet drop (PD), for dedicated short-range communications (DSRC). The dataset used for training, validating, and testing was extracted from experiments under several different road scenarios including highways, local areas, residential areas, state parks, and rural areas. The study shows that the proposed PL prediction model outperforms conventional empirical models. Meanwhile, the proposed PD prediction model achieves higher prediction accuracy than the statistical one. Moreover, the prediction model can operate in real-time, through updating its training set, to predict channel parameters. Such a model can be easily extended to the applications of autonomous driving, the Internet of Things (IoT), 5th generation cellular network technology (5G) and many others.

## 1. Introduction

### 1.1. Background and Motivation

Channel modeling is essential for the design and evaluation of the protocols adopted in vehicular communication and networks [1]. For vehicular wireless communication, if the sensitivity of the receiver is constant, the state of the transmitter and the path loss (PL) will become the main factors affecting the communication transmission quality [2]. In reality the road environment is complex: Conditions such as occlusion, reflection, multi-path so on and so forth, can have a negative impact on PL. Therefore, predicting PL in time and then making corresponding compensation adjustments at the transmitter can effectively guarantee the link quality. In addition, in some cases the transmission environment is too complicated. This leads to the fact that, based on the prediction of the PL, even if the transmitter operates in the limit state the communication may still be unstable. In this case, it is necessary to effectively predict the packet drop rate (PDR) and then adjust the transmission strategy. According to the prediction result of the PDR, before the environment further deteriorates the data with a low packet loss rate are preferentially transmitted.

Vehicle-to-vehicle channels differ significantly from those of cellular systems due to time varying environments that consist of roads, surrounding vehicles, and objects [3]. There are two main types of channel prediction models:
⬤**Theoretical model:**Theoretical models mainly include ray optical prediction models such as ray-tracing models [4,5] and ray-launching models [6,7]. Ray tracing models are approximations of Maxwell equations and are based on simulations. These models can accurately depict the channel. However, they suffer from heavy computational load; therefore, it is difficult to use them to depict the variance of the real world in real-time.⬤**Empirical model:**Empirical models, such as the dual-slope (DS) distance-breakpoint PL [8,9], polynomial fitting [10], and generalized gamma (GG) shadowed fading models [11], could be used as general reference models, but are not appropriate for real-time applications.

This paper proposes a real-time neural network (NN) model to predict channel parameters based on the latest data collected. This NN model has the ability to self-train and adapt to changes in the environment or the scene. Therefore, it is suitable for real-time channel prediction for vehicular communications. In addition, unlike empirical models, the NN model can fit general distributions while retaining local details. Although the NN approach has been widely researched in recent years, its application in communication systems, especially channel prediction, is still in its infant stages [12,13].

### 1.2. Related Work

Prediction of channel parameters, including PL and packet drop (PD), is the foremost task in channel modeling [14]. Traditionally, communications systems are modeled by mathematical superposition, approximation, and fitting, which suffer from the oversimplified real-world issues. NNs have overcome these issues and have been increasingly utilized in communications recently.

The proposed PL and PD NN-based prediction model reflects the relationship between the input (distance, velocity, etc.) and the output (PL, PD, etc.) in connectionism under various environments.

Replacing the traditional communication modules with multilayer NN models is a new research topic. Thanks to the update of NN algorithms and upgrades of computer hardware (e.g., the graphics processing unit—GPU) over recent years, the application of NNs for communications has been studied. In the field of wireless communication, deep learning has been used for modulation detection [15], channel encoding and decoding [16,17], and channel estimation [18,19]. A novel iterative belief propagation-convolutional neural network (BP-CNN) architecture has been proposed for channel decoding under correlated noise. Iterating between BP and CNN will gradually improve the decoding signal-to-noise ratio (SNR) and, hence, result in a better decoding performance [17]. Similarly, a channel estimation and signal detection approach based on deep learning for orthogonal frequency-division multiplexing (OFDM) systems is proposed in [18]. Unlike the current least squares (LS) and minimum mean-square error (MMSE) estimation methods, it estimates channel state information (CSI) implicitly and thus recovers the transmitted symbols directly. 

Furthermore, NNs can replace the entire blocks in the communication system with the same or even better performance [20]. The communication system is interpreted as an auto-encoder for the first time in [15]. Communications system design is considered as an end-to-end reconstruction task that seeks to jointly optimize transmitter and receiver components in a single process by deep learning. After that, a complete communications system solely composed of NNs using unsynchronized off-the-shelf software-defined radios and open-source deep-learning software libraries was built in [21].

As for PL models, various methods are applied in PL models, such as theoretical models based on Maxwell’s equations, ray tracing models based on simulation, and empirical models, parameters of which are derived from experimental data for each defined category. Empirical models characterize the channels in corresponding scenarios well. The typical free space channel model and Stanford University Interim (SUI) models were applied to urban environments in centimeter-wave and millimeter-wave frequency bands [22]. Improved Rural Macrocell (RMa) models were applied to rural areas in the frequency range from 0.5 GHz to 30 GHz [23]. A bounded model is used to calculate the decay of signal power caused by PL [24]. Artificial neural network (ANN) models were used for macrocell PL prediction [25], and a 3rd Generation Partnership Project (3GPP) blockage model was utilized to predict the PL caused by the blockage of traffic sign posts [26]. When the environment changes, these kinds of empirical models are no longer valid. An Alpha-Beta-Gamma (ABG) model and a Close-In (CI) free-space reference distance model was introduced to predict PL through a measured dataset [27]; these are also empirical models but can adapt to changes in the environment. In addition, there are many NN-based PL methods in wireless communications [28,29,30,31]. They are very valuable and useful in static PL prediction. However, in real-time PL prediction, we need to predict the real-time PL in a short time. That is, these methods face the drawback of sacrificing local information or over-fitting when predicting real-time PL in wireless communication. In contrast, the proposed NN-based PL model can overcome these deficiencies and show more accurate results in real-time PL prediction.

In wireless communication, the performance of the system will be affected when the PDR or round-trip latency increase [32]. The packet with excessive delay is usually discarded to achieve the minimum and balanced PDR for wireless traffic [33]. The research included a PD algorithm for various packet loss rates [34], improved algorithm for latency and PD [35], and PDR optimization algorithm [36]. Regarding PD, most of the related research is focused on controlling unreliable communication channels caused by known PDs without any predicted information to enhance the system performance [37,38,39]. This paper presents a solution to predict PD which can be utilized to improve system reliability.

In the field of vehicular communications, there is scant relevant research on a unified or a general channel prediction model, of which the NN structure to predict PL and PD are similar. The prediction model updates its training set with most recent measurements in real-time. However, the proposed NN based model takes multiple factors into account. Thus, it could model local situations in a higher granularity than traditional ones.

### 1.3. Contributions and Innovations

The main contributions of this paper are summarized as follows:
⬤The proposed NN model can be used for predicting PL. The PL prediction model shows better performance compared to that of commonly used empirical models, such as the dual-slope distance-breakpoint PL model, polynomial fitting model, and GG shadowed fading model.⬤The proposed NN model can be used to predict PL in real-time and it can be updated in real-time. Therefore, the model not only guarantees the real-time performance, but also updates the PL prediction locally according to the actual environment with higher accuracy.⬤The proposed NN model can be used for predicting PD. The model predicts the PDR in the future period by the PD situation in the current time slot. And, with the acquisition of new data, the PD prediction model is updated in real-time. This model provides a higher prediction accuracy than the statistics-based model.

The innovations of this paper are reflected in the following two aspects. 

⬤In the theoretical aspect, the paper introduces the ANN algorithm into the field of vehicular communication. Although artificial neural algorithms are making rapid progress, this is an innovative application for vehicular communication, in terms of channel modeling and prediction.⬤In the semi-physical simulation, this paper proposes a prediction model based on ANN. The model can predict channel parameters, including PL and PDR, based on the measured dataset. The simulation results show that: (1) in the PL offline prediction part, the ANN model is more accurate than the traditional model; (2) in the PL online prediction part, the ANN model has a better ability to fit the dataset than the traditional model, and is more capable of reflecting the details of different time steps; and (3) in the PDR online prediction part, the ANN model has a better trade-off between precision and accuracy than the traditional model.

The remainder of this paper is organized as follows. The second section presents how to obtain data from dedicated short-range communications (DSRC) on-board units (OBUs) such as global positioning system (GPS), speed, and received signal strength indicator (RSSI). Section 3 introduces a proposed model to predict PL in real-time. This NN-based model has a better ability to predict the channel PL distribution than the three current empirical models. Moreover, this model can update the channel predictions in real-time. In Section 4, we use the same model for real-time PD prediction. The proposed NN-based PD prediction model provides lower mean square error (MSE) and higher accuracy than the statistics-based model.

## 2. DSRC Network Sniffer and Data Connection

The experiments were carried out by two midsize vehicles, and each roof mounts a DSRC OBU at an approximate height of 1.48 meters. The OBUs were outside of the vehicles for most of the tests. However, some tests were performed with OBUs placed inside the vehicles due to heavy rain to prevent the electronics getting wet. The GPS antennas had to be mounted outside of the vehicles in all weather conditions; otherwise, the GPS position would stop updating. An additional micro-controller (Raspberry Pi) was connected with each of the DSRC OBUs via a RS-232 serial interface for data exchange. A diagram of the DSRC network sniffer and data connection is shown in Figure 1.

We developed the software by using Autotalks SDK (Software Development Kit) for device configuration and data generation running on the DSRC OBUs. The data include the current longitude, latitude, altitude, heading angle and ground speed from the GPS sensor, as well as the timestamps, packet counters, and RSSI of each packet. The timestamps are inserted before each packet has been sent and after it has been received on both OBUs. Packet counters were used for PD rate calculation. On the micro-controllers, we developed a GUI (graphical user interface) based on Linux OS to collect data from the OBUs as well as to send the configuration parameters to control them. The collected data can be stored locally or sent to our private server for real-time display. The custom parameters include data rate, packet rate, packet size, transmit power, channel number, priority, etc. A DSRC OBU, GPS antenna, and an additional micro-controller for the DSRC network vehicle experiments, are shown in Figure 2.

All experimental data were collected with the same device configurations. We set the transmit power at 23 dBm, the data rate at 6 Mbps, 1000 bytes packet size, and 10 Hz packet rate. Channel 172 and 174 were used for the bidirectional communication. The band of the DSRC was allocated at 75 MHz of spectrum in 5.9 GHz. There are 7 channels in total, each occupying 10 MHz of bandwidth. The center frequencies of channel 172 and channel 174 were 5.86 GHz and 5.87 GHz, respectively.

Parameter $pc_{Tt}$ is the packet counter of the current packet and $pc_{Tt-1}$ is the packet counter of the previous packet. If $pc_{Tt}-pc_{Tt-1}>1$, we call it a PD event, and the count of lost packets is recorded as $c_{t}=pc_{Tt}-pc_{Tt-1}-1$. The round-trip PDR can be expressed by the total number of packets drop per M packets. Round-trip PDR can be easily calculated through
(1)$${\mathrm{PDR}}_{\mathrm{round}-\mathrm{trip}}=(\frac{1}{M}\sum_{t=1}^{M}c_{t})\cdot 100\%.$$

The distance d between two vehicles can be calculated by GPS information:
(2)$$\begin{array}{l} a=\sin^{2}(\Delta \varphi /2)+\cos\varphi_{1}\cdot \cos\varphi_{2}\cdot \sin^{2}(\Delta \lambda /2)\\ c=2a\tan2(\sqrt{a},\sqrt{1-a})\\ d=R\cdot c \end{array},$$where φ is latitude, λ is longitude, and R is earth’s radius (mean radius = 6371 km).

The measurements were conducted in Mcity Ann Arbor, UM-D Campus, and cities such as Ann Arbor, Dearborn, and Detroit. Weather conditions included rain, sun and showers. Most common road scenarios were covered, including the following:
⬤Highways⬤Local areas⬤Residential areas⬤State Park and rural areas

In order to simulate all the common traffic scenarios, two testing vehicles were driven in various patterns throughout the experiments. These included following, overtaking, lane changing, opposite driving, intersection scenarios with traffic lights, and stop signs. Multiple experiments were conducted during both peak and off-peak hours with a maximum driving speed of 40 km/h in residential areas, 70 km/h in Mcity and local areas, and 120 km/h on highways. The distance between the two vehicles was relatively close, ranging from 3 to 120 m. Apart from the speed differences, there were some unique characteristics that came from those different environments. For instance, campus provided the driving scenarios with pedestrians, state park and rural areas provided an open field environment, and Detroit downtown represented the urban scenario with crowded traffic.

Figure 3 shows one of our experiments in Ann Arbor. The relative distance was calculated using the GPS data for the two vehicles and the relative speed is the difference between two speed vectors. In the following sections, we will use the dataset obtained from the experiments to build empirical models and an NN model to predict channel parameters such as PL and PDR.

## 3. A Novel Path Loss Prediction Model

This section proposes an NN-based PL prediction model trained by a BP algorithm, which corresponds to our first contribution. In the second subsection, this paper expands the offline prediction model to a real-time NN-based PL prediction model, which is our second contribution. 

For wireless communications systems, PL prediction is fundamental for system planning and optimization. A general PL model is usually specified to categorized environments, such as highways, local areas, residential areas and rural areas, by assigning the maximum, minimum, median, mean and standard deviation of PL, which are shown in Table 1.

DSRC channels are time-varying and highly environment independent, resulting in a great number of channel parameters listed above. The NN-based PL model presented in this paper is general and applicable to diverse scenarios without changing the configuration and weights of NN. Moreover, simulation results indicate that it has lower root mean square error (RMSE) and higher accuracy than conventional models, as shown in Figure 4.

### 3.1. Typical Empirical Models for Path Loss Prediction

Compared to theoretical modes, empirical models are built upon measured data featuring flexible and accurate prediction performance without requiring prior knowledge. Three empirical models are discussed for comparison purposes in the following: (1) Dual-slope distance-breakpoint PL model; (2) polynomial fitting model, and (3) GG shadowed fading model.

#### 3.1.1. Dual-Slope Distance-Breakpoint Path Loss Model

In a large-scale propagation model, PL is only related to the distance between the transmitter and receiver for a given frequency. Two PL indexes, $n_{1}$ and $n_{2}$, are adopted to described the fading slopes in two distinguished regions separated by Fresnel distance. PL index and the variance of slow fading or shadowing can be calculated from measured data using the MMSE criterion. The PL at d, the distance between the Tx and Rx, can be expressed as
(3)$$PL_{DS}(d)[dB]=PL(d_{0})+\{\begin{array}{cc} 10n_{1}\log10(\frac{d}{d_{0}})+X_{\sigma 1} & d_{0}\leq d\leq d_{b}\\ 10n_{1}\log10(\frac{d_{b}}{d_{0}})+10n_{2}\log10(\frac{d}{d_{b}})+X_{\sigma 2} & d>d_{b} \end{array},$$where $X_{\sigma 1}$ and $X_{\sigma 2}$ are the variances of slow fading, and $PL(d_{0})$ the PL at the reference distance $d_{0}$.

(4)$$PL(d_{0})=32.45+20\log_{10}(\bar{d_{0}})+20\log_{10}(\bar{f})-G_{t}-G_{r}.$$

Among them, $\bar{d_{0}}$ and $\bar{f}$ are normalized in km and MHz, respectively.

In conventional Line of sight (LoS) models, $d_{b}$ is the Fresnel distance to the point where the first Fresnel zone touches the ground, which can be calculated by $d_{b}=\frac{4h_{T}h_{R}}{\lambda }$, where $h_{T}$ and $h_{R}$ are antenna heights of the transmitter and receiver, respectively, and λ is the wavelength [9].

The PL model is built up through the steps described below.
(i)The relative distances in the training set are rounded to the nearest meter, denoted as $d_{k}$, k={5,6,⋯,199}. The corresponding PLs are denoted as $PL(d_{k})=\{PL{\left(d_{k}\right)}_{1},PL{\left(d_{k}\right)}_{2},\cdots \}$.(ii)Take $d_{0}=\{d_{0_{1}},d_{0_{2}},\cdots \}$ into Equation (4) to calculate the PL at the reference distance $PL(d_{0})$.(iii)Calculate the average of PLs at each relative distance $d_{k}$, denoted as $E\{PL(d_{k})\}$.(iv)For each reference distance $d_{0_{p}}$, take $n_{1}=\{0,0.01,0.02,\cdots ,10\}$ into Equation (3) (subject to $d_{0}\leq d\leq d_{b}$) to calculate the optimal $n_{opt1}$ and $X_{\sigma 1}$, which leads to the minimum RMSE of $E\{PL(d_{k})\}$ and $PL_{DS}(d_{k})$.(v)For each reference distance $d_{0}{}_{p}$, take $n_{2}=\{0,0.01,0.02,\cdots ,10\}$ into Equation (3) (subject to $d>d_{b}$) to calculate the optimal $n_{opt2}$ and $X_{\sigma 2}$, which leads to the minimum RMSE of $E\{PL(d_{k})\}$ and $PL_{DS}(d_{k})$.

In the PL prediction simulation using the FSPL-based model, six reference values $d_{0}=\{5,6,\cdots ,10\}m$ and corresponding PL indexes of $n_{1}$ and $n_{2}$ were calculated. The predicted MSE of the prediction models using a training set, validation set, and test set were calculated. Simulation results show that, regardless of the reference distance, the NN-based model outperforms the FSPL-based model in terms of prediction accuracy.

#### 3.1.2. Polynomial Fitting Model

Polynomial fitting model (PFM) is another commonly used model based on measured data through the least-squares polynomial curve fitting. An approximate curve described by the function of y=f(x) is drawn from the experimental data where the curve does not need to pass through modeling data exactly. The expression of the approximate curve is
(5)$$y=\varphi (x)=\sum_{j=0}^{k}a_{j}x^{j},$$where k is the order of fitting curve, which is selected according to the loss function of the sum of the squares of the errors. It is then adapted to the above polynomial equation. The sum of the distances from each sample point to the fitted curve, representing the sum of squared deviation, is calculated by

(6)$$E=\frac{1}{2}\sum_{i=1}^{n}{\left(y_{i}-\sum_{j=0}^{k}a_{j}x_{i}^{j}\right)}^{2}.$$

The value of $a_{j}$ that minimizes E satisfies the condition that $\partial E/\partial a_{j}=0$. After simplifying and using the matrix representation, one can have

(7)$$\left[\begin{array}{cccc} 1 & x_{1} & \cdots & x_{1}^{k}\\ 1 & x_{2} & \cdots & x_{2}^{k}\\ \vdots & \vdots & \ddots & \vdots \\ 1 & x_{n} & \cdots & x_{n}^{k} \end{array}\right]\left[\begin{array}{c} a_{0}\\ a_{1}\\ \vdots \\ a_{k} \end{array}\right]=\left[\begin{array}{c} y_{1}\\ y_{2}\\ \vdots \\ y_{n} \end{array}\right].$$

Equation (7) can be further abbreviated as XA=Y and the fitting curve can be acquired by $A=X^{-1}Y$.

The polynomial model was established through the following steps.
(i)The relative distances in the training set are rounded to the nearest meter, denoted as $d_{k}$, k={5,6,⋯,199}, while corresponding PLs are denoted as $PL(d_{k})=\{PL{\left(d_{k}\right)}_{1},PL{\left(d_{k}\right)}_{2},\cdots \}$.(ii)Calculate the average of PLs at each relative distance $d_{k}$, denoted as $E\{PL(d_{k})\}$.(iii)Take $k=\{k_{1},k_{2},\cdots \}$ into Equation (5) to calculate the approximate curve for order $k_{p}$.(iv)Calculate the coefficient matrix of $A_{p}=X^{-1}Y$ using Equation (7) according to the order of $k_{p}$.(v)For each order $k_{p}$, take $A_{p}$ into Equation (5) to calculate the optimal $k_{opt}$, which generates the minimum RMSE of $E\{PL(d_{k})\}$ and $PL_{PF}(d_{k})$.

In the simulation of the PFM, fitting orders of k={1,2,⋯,24} were picked, where 24 was selected by Monte Carlo simulation. The predicted MSE of the PFM and NN models based on a training set, validation set, and test set was then calculated. Simulation results validate that for all orders tested, the NN-based model has lower MSE, that is, more accuracy of prediction when compared with the PFM-based model.

#### 3.1.3. Generalized Gamma Shadowed Fading Model

Vehicular channels may experience fast fading as well as shadowing simultaneously. A model that reflects the two types of fading is the GG distribution, of which the Probability Distribution Function (PDF) and Cumulative Distribution Function (CDF) can be written as
(8)$$\begin{array}{l} f_{GG}(p_{r})=\frac{s}{\Gamma (m)P_{g}{}^{m}}p_{r}{}^{ms-1}e^{-\frac{p_{r}{}^{s}}{P_{g}}}\\ F_{GG}(p_{r})=\frac{s}{\Gamma (m)P_{g}{}^{m}}\int p_{r}{}^{ms-1}e^{-\frac{p_{r}{}^{s}}{P_{g}}}dp_{r} \end{array},$$where m is the fading parameter, s the shape parameter (typically restricted to 0<s≤1), and Γ(⋅) the gamma function. The average power of GG channel, $P_{g}$, is expressed as
(9)$$P_{g}={\left(\frac{\omega \Gamma (m)}{\Gamma (m+\frac{1}{s})}\right)}^{s},$$where ω is the time average of received power. A measurement study shows that m and s vary with distance [3]. 

The GG PL model is built up through the steps described below.
(i)The relative distances in the training set are rounded to the nearest 5 m, denoted as $d_{k}$, k={5,10,⋯,195}. The corresponding PLs are denoted as $p_{r}(d_{k})=\{p_{r}{\left(d_{k}\right)}_{1},p_{r}{\left(d_{k}\right)}_{2},\cdots \}$.(ii)Calculate PDF $f(p_{r})$ at the relative distance $d_{k}$ of the received signal power.(iii)Take m={0.5,0.6,⋯,20} and s={0.1,0.2,⋯,1} into Equation (8) to calculate the optimal $m_{opt}(d_{k})$ and $s_{opt}(d_{k})$, which lead to the minimum RMSE of $f(p_{r})$ and $f_{GG}(p_{r}:m,s)$.(iv)Putting $m_{opt}(d_{k})$ and $s_{opt}(d_{k})$ into Equation (8), we get the estimated PDF and CDF of the received signal power at the relative distance $d_{k}$, denoted as ${\hat{f}}_{GG}(p_{r}:m_{opt}(d_{k}))$ and ${\hat{F}}_{GG}(p_{r}:m_{opt}(d_{k}))$, respectively.(v)The estimated received signal power at the relative distance $d_{k}$ is ${\hat{p_{r}}}_{GG}(d_{k})=\sum p_{r}(d_{k})\cdot {\hat{F}}_{GG}(p_{r}:m_{opt}(d_{k}),s_{opt}(d_{k}))$.

### 3.2. A Novel Neural Network-Based Path Loss Prediction Model

In this subsection, an NN-based PL prediction model is discussed. The input of the NN is the distance between Tx and Rx, gained from onboard GPS sensors, while the PL was calculated from RSSI at Rx. All measurement data came from the above experiments.

The weight of the path from the $i_{th}$ input layer to the $j_{th}$ hidden layer is denoted as $w_{ij}$ and the weight from the $j_{th}$ hidden layer to the $k_{th}$ output layer is $w_{jk}$. The number of nodes in the input layer, hidden layer, and output layer were $N_{1}$, $N_{2}$, and $N_{3}$, respectively. Thus, we have $i=\{1,2,\ldots ,N_{1}\}$, $j=\{1,2,\ldots ,N_{2}\}$, and $k=\{1,2,\ldots ,N_{3}\}$. The weight vectors of each perceptron were optimized using the back-propagation algorithm. We used the accumulated error BP algorithm and adjusted $w_{ij}$ and $w_{jk}$ to minimize the global error E. NN computations, starting from input data and going on to the hidden/output layer weights computations, are shown in Table 2.

Pseudo-code of NN development is presented as the following.
**Algorithm 1** Developed of Neural Network**Input:**1: $dataset=[train[d_{k},PL(d_{k})],validation[d_{k},PL(d_{k})],test[d_{k},PL(d_{k})]]$2: $para_{hyper}$ = [] // NN hyper-parameters including number of hidden layers, nodes of each hidden layers, activation function, learning rate, epochs, training function, learning function, etc.**Output:**1: pred // Predict $PL_{NN}(d_{k})$ at each relative distance $d_{k}$2: rmse //RMSE of $PL(d_{k})$ and $PL_{NN}(d_{k})$3: ACC // Accuracy of $PL(d_{k})$ and $PL_{NN}(d_{k})$**for**$para_{hyper}^{i}$ in $para_{hyper}$
**do**$net=train\_validation(train,\;validation,para_{hyper}^{i})$$pred=net(test,\;para_{hyper}^{i})$E=RMSE(pred,test)**end for**$para_{hyper}^{opt}=\arg\min(E)$$pred^{opt}=net(test,para_{hyper}^{opt})$$rmse^{opt}=RMSE(pred^{opt},test)$$ACC^{opt}=Accuracy(pred^{opt},test)$**return**$pred^{opt}$, $rmse^{opt}$, $ACC^{opt}$

With the NN developed in Algorithm 1, the prediction model is built up through the steps described below.
(i)The relative distances in the training set are rounded to the nearest meter, denoted as $d_{k}$, k={5,6,⋯,199}. The corresponding PLs are denoted as $PL(d_{k})=\{PL{\left(d_{k}\right)}_{1},PL{\left(d_{k}\right)}_{2},\cdots \}$.(ii)Randomly divide the dataset into training set, validation set, and test set in proportions of 60%, 15%, and 25%, respectively.(iii)Set NN parameters: number of nodes in each layer $N_{1}=1,N_{2}=24,N_{3}=1$, activation function tanh, learning rate η=0.05, maximum number of epochs to train epochs=60, sum-squared error goal goal=0.001, back-propagation weight/bias learning function learngdm, etc.(iv)Train the NN model with the training set, and then tune the model’s hyper-parameters with the validation set to reduce overfitting.(v)Predict $PL_{NN}(d_{k})$ at each relative distance $d_{k}$ with the NN model from (iv) and then calculate the RMSE of $PL(d_{k})$ and $PL_{NN}(d_{k})$.(vi)Repeat step (iii)–(v) when the scenario changes to obtain the optimal combination of parameters. In the same scenario, the NN parameters are fixed after the optimal parameters are obtained.(vii)Estimate $PL_{NN}(d_{k})$ at each relative distance $d_{k}$ with The NN model from (vi) and then calculate the RMSE of $PL(d_{k})$ and $PL_{NN}(d_{k})$.

It can be seen clearly from Figure 4 that the NN model has the lowest MSE, that is, the best accuracy among models. Simulations covered four road scenarios including rural areas, residential areas, local areas, and highways.

The error bars in Figure 4 show the relationship between distance and PL, where distances were rounded appropriately depending on the measured dataset. Figure 4 also illustrates the mean of μ and the standard deviation of σ of PL at each distance. Table 3, Table 4, Table 5 and Table 6 indicate statistical characteristics, such as maximum, minimum, mean, and standard deviation, of the measurement data, that is, the dataset used for PL prediction.

The orange dashed line was calculated from the dual-slope model. The breakpoint was set as $d_{b}$ = 171.4353 m. The minimum reference distance was set as $d_{0min}$ and $d_{0}=\{d_{0min},d_{0min}+1,\cdots ,d_{0min}+6\}$. Among them, the smallest RMSE, distances and PL indexes are all shown in Figure 4. As one can see from Figure 4, the prediction curve of the dual-slope model fits the average of the dataset and is always located within [μ−σ,μ+σ].

The yellow dash-dot line is gained from the PFM. Cases where the order was k={1,2⋯,24} are simulated. Simulation results in Figure 3 show that, generally speaking, PFM performs better when *k* = 1 or *k* = 2. The prediction curves were mostly distributed within the range of [μ+σ,μ+2σ], which is not centered around μ and is not able to predict the PL accurately.

The purple dotted line was calculated from the GG model. The prediction curve fluctuates, reflecting the trend of PL but in smaller prediction values.

The blue line labelled with triangles was gained from the NN prediction model. The results from the NN model outperform those from the other three models. The prediction curve fits the average of the dataset and always remains within in [μ−σ,μ+σ]. For all road scenarios, it shows both the trend and the details of PL. The peaks, such as at 80–90 m in the rural case and at 90–100 m in the local case, etc., were likely caused by up fading, that is, multi-path conditions causing constructive summation of several multi-path components. 

For the four scenarios, the corresponding modeling parameters vary.
⬤In the Local case, the optimal parameters of the dual-slope model are $d_{0}=$ 15 m, $n_{1}=$ 1.34, $n_{2}=$ 1.36, and that of the PF model is order= 1.⬤In the Residence case, the optimal parameters of the dual-slope model are $d_{0}=$ 10 m, $n_{1}=$ 1.22, $n_{2}=$ 1.48, and that of the PF model is order= 1.⬤In the Rural case, the optimal parameters of the dual-slope model are $d_{0}=$ 12 m, $n_{1}=$ 1.64, $n_{2}=$ 2, and that of the PF model is order= 2.⬤In the Highway case, the optimal parameters of the dual-slope model are $d_{0}=$ 54 m, $n_{1}=$ 1.88, $n_{2}=$ 2.07, and that of the PF model is order= 1.

As shown above, the NN PL model achieves the best performance because the NN line is closer to the real data than the other estimations.

The DS breakpoint model is limited by using log(⋅) to approximate the trend. High-order PFM has an accuracy problem at the extremes of the range which results in reduced accuracy. The PF-based model sacrifices accuracy slightly due to the continuity and smoothness of the mathematical model adopted. Moreover, the PF-based model degraded extremely at the edge of the data range. The GG model has more accurate predictions than those from the PF model based on a large dataset, which is not suitable for the proposed real-time PL prediction. In addition, to find the optimal values of m and s, all training sets are iterated repeatedly. The number of iterations is positively correlated with the accuracies of m and s. Such processing takes time, which is not acceptable for real-time operation.

However, the NN-based PL prediction model uses a new way to explain relationships within the dataset, which is more in line with the actual channel distribution. It can predict channel parameters for hardware-in-the-loop simulations and modeling for vehicle networks, autonomous driving and networked controlled robotics.

### 3.3. A Novel Real-Time Neural Network-Based Path Loss Prediction Model

For the applications of vehicle communications and autonomous driving, the channel is time-varying. During the training phase, outdated data will be dropped as appropriate while updating the data.

The foremost task is to identify the number of packets to be used for a training set in a real-time model. Simulation results showed that the number of packets is suggested to be no less than 900. In each prediction period, 900 packets were therefore divided into training set, validation set, and test set in proportions of 60%, 15%, and 25%, respectively. 

Window factor, noted by s, is the percentage of the dataset that is updated. Datasets were updated chronologically while the window factor was selected as s=5%. The time window, noted by $t_{s}$, is given by
(10)$$t_{s}=l_{data}\cdot s\cdot t_{packet},$$where, $l_{data}$ is the number of packets of the dataset and $t_{packet}$ the duration of the packet.

The time window can be reduced by shortening the time duration of the packet if necessary. 

Figure 5 compares the offline and real-time PL prediction model based on the NN. In the offline model, the whole dataset is used, while the real-time model only uses data that is within the coherent window described above.

The coverage areas are different in each slot. The PL prediction for both offline (orange *) and real-time models (colored lines) generated similar results. Figure 6 shows the correlation coefficients of 6 real-time PL prediction curves. The dataset adopted is from the rural areas, but can be applicable to others as well. Different areas lead to individual PL curves, but all function well. For simplicity, the following results are based on rural areas only.

The real-time PL prediction curves are defined as $\hat{PL_{i}}$ and $\hat{PL_{j}}$ in different slots, where i={1,2,…,6} and j={1,2,…,6} in this case. The correlations of PL curves are meaningful and defined as $\hat{PL_{i}^{\prime }}$ and $\hat{PL_{j}^{\prime }}$. The correlation coefficient is used as a measure of the linear dependence between them. If a variable has $N_{ij}$ scalar observations, then the Pearson correlation coefficient is defined as
(11)$$\rho (\hat{PL_{i}^{\prime }},\hat{PL_{j}^{\prime }})=\frac{1}{N_{ij}-1}\sum_{p=1}^{N_{ij}}{\left(\frac{{\hat{PL_{i}^{\prime }}}_{p}-\mu_{i}}{\sigma_{i}}\right)}^{T}(\frac{{\hat{PL_{j}^{\prime }}}_{p}-\mu_{j}}{\sigma_{j}}),$$where $\mu_{i}$ and $\sigma_{i}$ are the means and standard deviations of $\hat{PL_{i}^{\prime }}$, respectively, and $\mu_{j}$ and $\sigma_{j}$ the means and standard deviations of $\hat{PL_{j}^{\prime }}$. 

The cross-correlation matrix can be expressed as

(12)$$\rho_{PL}=\left[\begin{array}{cccc} \rho (\hat{PL_{1}^{\prime }},\hat{PL_{1}^{\prime }}) & \rho (\hat{PL_{1}^{\prime }},\hat{PL_{2}^{\prime }}) & \cdots & \rho (\hat{PL_{1}^{\prime }},\hat{PL_{6}^{\prime }})\\ \rho (\hat{PL_{2}^{\prime }},\hat{PL_{1}^{\prime }}) & \rho (\hat{PL_{2}^{\prime }},\hat{PL_{2}^{\prime }}) & \cdots & \rho (\hat{PL_{2}^{\prime }},\hat{PL_{6}^{\prime }})\\ \vdots & \vdots & \ddots & \vdots \\ \rho (\hat{PL_{6}^{\prime }},\hat{PL_{1}^{\prime }}) & \rho (\hat{PL_{6}^{\prime }},\hat{PL_{2}^{\prime }}) & \cdots & \rho (\hat{PL_{6}^{\prime }},\hat{PL_{6}^{\prime }}) \end{array}\right].$$

The theory of strict frames (rules) for correlation may be presented as follows:
⬤0≤|ρ|≤0.20: the correlation is non-important;⬤0.20<|ρ|≤0.60: the correlation is weak;⬤0.60<|ρ|≤0.90: the correlation is strong;⬤0.90<|ρ|≤1.00: the correlation is very strong.

The correlation coefficients of two consecutive slots are close to one. This result is in line with expectations:
⬤The updated dataset can effectively predict the trend of PL. The PL prediction curves at different slots have coherent shapes and trends at the same distance.⬤It also varies practically as the environment changes. In some cases, even though Tx and Rx take the same distance, the PL shape predicted demonstrated variations. This is due to changes in the environment where the two vehicles are located, such as obstacles and occlusion.

The real-time PL prediction model was proven to be able to predict not only the trend, but also the local variations in real time.

## 4. A Real-Time Packet Drop Prediction Model

This section proposes a real-time NN based PDR prediction model which corresponds to our third contribution. Besides PL, there is another parameter in wireless communication, that is, PD. The proposed NN model for predicting channel parameters is applicable not only to PL but also to other parameters. Therefore, in this section, we will present the performance of the proposed NN prediction model for the channel parameters, that is, PD. It has been found that PD is partially relevant to relative distance, relative speed, PL, and frequency. Many causes are hard to describe with a mathematic model, let alone the real-time implementations. To this end, this section proposes a real-time prediction model of PD probability based on NNs. 

Packet drop series could be regarded as time series, which could be predicted by both the NN-based method and statistical average method. The proposed model takes $n_{1}$ packets to predict PDR in following $n_{2}$ packets. The model updates itself in every $n_{3}$ packets. Usually, $n_{3}<n_{2}$; thus, the prediction window is able to cover transmission times of interest. The framework of the PDR prediction is shown in Figure 7.

In the simulation, the dataset is randomly selected from the above experiments. The dataset is divided into training, validation and test sets according to the proportions of 60%, 15%, and 25%, respectively. The input is either 0 or 1, representing whether the packet is received (input = 0) or dropped (input = 1). The label is the round-trip PDR in the next $n_{2}$ packets. The structure of the PRD prediction neural network is shown in Figure 8. The activation function is f(⋅)=tanh(⋅) and the learning rate is η=0.05. The number of neurons in the hidden layers are $N_{2}^{\left(1\right)}=150$, $N_{2}^{\left(2\right)}=100$, $N_{2}^{\left(3\right)}=50$, and $N_{2}^{\left(4\right)}=25$, respectively. The outputs of the four hidden layers are

(13)$$\begin{array}{cc} H_{j}^{\left(1\right)}=f(\sum_{i=1}^{N_{1}}w_{ij}x_{i}+b_{j}^{\left(1\right)}) & H_{m}^{\left(2\right)}=f(\sum_{j=1}^{N_{2}^{\left(1\right)}}w_{jm}H_{j}^{\left(1\right)}+b_{m}^{\left(2\right)})\\ H_{n}^{\left(3\right)}=f(\sum_{m=1}^{N_{2}^{\left(2\right)}}w_{mn}H_{m}^{\left(2\right)}+b_{n}^{\left(3\right)}) & H_{p}^{\left(4\right)}=f(\sum_{n=1}^{N_{2}^{\left(3\right)}}w_{np}H_{n}^{\left(3\right)}+b_{p}^{\left(4\right)}) \end{array}.$$

Simulations were performed to validate the effectiveness of the real-time PD prediction model under different values of $n_{1}$, $n_{2}$, and $n_{3}$, with the criteria described as below:
⬤The value of $n_{1}$ should be carefully chosen to balance between past data and prediction data. In the model, $n_{1}=\{100,120,\cdots ,500\}$.⬤The value of $n_{2}$ should be selected in such a way that corresponding prediction time should be within the channel coherent time. In the model, $n_{2}=\{2,4,\cdots ,30\}$.⬤$n_{3}$ is selected as to be equal or less than v resulting in a prediction window covering transmission times of interest. In the model, $n_{3}=0.5\cdot n_{2}$, i.e. $n_{3}=\{1,2,\cdots ,15\}$.⬤Several combinations of $n_{1}$, $n_{2}$, and $n_{3}$ were adopted in the simulation to get a good prediction accuracy. A comparison factor is defined as, $\alpha =accuracy/MSE^{2}$. The set of $n_{1}$, $n_{2}$, and $n_{3}$ resulting in the largest α is optimized.

The simulation results are shown in Figure 9. It shows the values of MSE during the training, validation, test, and overall set. MSE roughly reduces with the decrease of $n_{1}$ or the increase of $n_{2}$. The set of $n_{1}$ = 180, $n_{2}$ = 20, and $n_{3}$ = 10 achieve the maximum value of α.

Next, the performance of the proposed NN based model was compared with that of the statistical average based model. The statistics-based model averages the values of PD in $n_{1}$ data to predict the probability of PD in the following $n_{2}$ data. Predictions are updated with every $n_{3}$ new data gained. Similarly, the NN-based model is constantly trained by the dataset, consisting of the inputs $n_{1}$ data, and outputs probability of PD in next $n_{2}$ data. With every $n_{3}$ new data gained, the prediction updates. 

Table 7, Table 8 and Table 9 list the PDF of the training, validation and test set. The probability of PD can be obtained through the softmax layer which are shown by bold style in Table 7, Table 8 and Table 9. 

According to the measured data, the probability of PD in 20 packages is $P_{PD}=\{0,0.05,\cdots ,0.6\}$. In Table 7, Table 8 and Table 9, each column lists a slot and the number of slots is determined by the length of dataset. Ten slots were selected to illustrate the results and principles of the PD prediction model. The probability corresponding to the maximum value of PDF in each slot is the output of the prediction model.

A comparison of the real-time prediction performance of the two models is shown in Figure 10. The prediction results are divided into four types: *A* indicates that the NN based model can predict the probability of PD, while a statistics-based model does not; *B* indicates that both models can predict; *C* implies that the statistics-based model can predict the probability of PD while the NN based model fails; and *D* shows that neither model can predict it. 

The RMSE and accuracy of real-time prediction models using the two methods is shown in Table 10. The accuracy of the NN-based model consists of case *A* and case *B* and the accuracy of a statistics-based model contains case *B* and case *C*. 

Simulation results show that the NN-based PD prediction model has a lower RMSE and higher accuracy than those of a statistics-based model. RMSE was reduced by 18.73% while the accuracy was increased by 85.19%. See Table 10 for more numerical comparisons.

The performance of the statistics-based model was not as good as the NN-based model because the dataset is not large enough. There are two contradictory conditions that limit the performance of the statistics-based model:
⬤Increasing the size of the dataset enhances the prediction accuracy of the statistics-based model.⬤Adding more datasets requires more previous time data, which are less relevant to the present and future slots. Furthermore, it reduces the accuracy of PD prediction.

The statistics-based model is, therefore, not applicable for wireless communications. In summary, the NN-based PD prediction model outperforms the statistics-based model and can be used to provide a good reference for the link status for wireless communications.

## 5. Conclusions

This paper proposes an NN-based PL and PD prediction model, working in either offline mode or online mode. The functionality has been well addressed and validated through extensive simulations. The NN-based channel prediction model is regarded as a new approach able to learn and predict both long-term and short-term variations of PL, PD and latencies. 

The proposed PL prediction model was carefully studied and compared with three typical PL prediction models, namely, dual-slope distance-breakpoint PL model, polynomial fitting model, and GG shadowed fading model. The study concludes that the proposed NN-based prediction model is has better performance when balancing between large-scale averaging and small-scale fitting.

Moreover, the proposed model features real-time implementation through updating its training dataset with the most recent measurements, while its transitional counterpart is incapable of doing this.

Furthermore, under the same conditions, the NN PD prediction model demonstrates better prediction capability if the appropriate time window is selected with enough coherent training sets. 

The next step of this effort is to present a novel channel prediction models based on long-short term memory (LSTM) to addresses the impact of various environmental parameters on channel parameters. LSTM could show more details among time series, compared with the BP algorithm utilized in this paper, through capturing both long term memory to reflect categories of environment adaptively, and short-term memory varying with consecutive channel prediction windows.

## Figures and Tables

**Figure 1 sensors-19-03541-f001:**
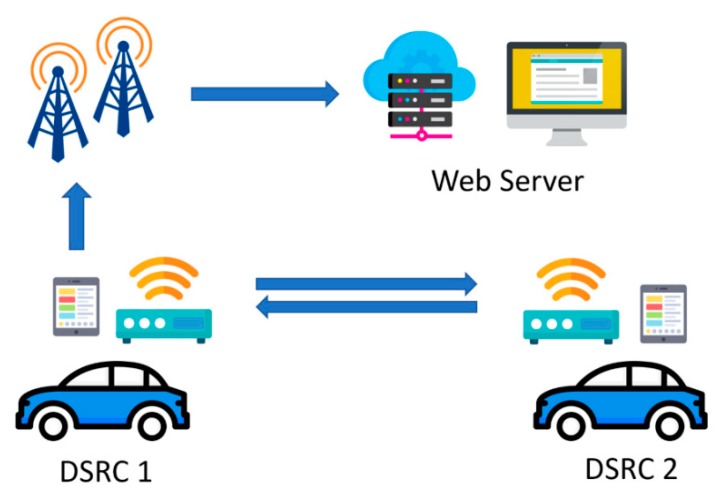
Diagram of the dedicated short-range communications (DSRC) network sniffer and data connection.

**Figure 2 sensors-19-03541-f002:**
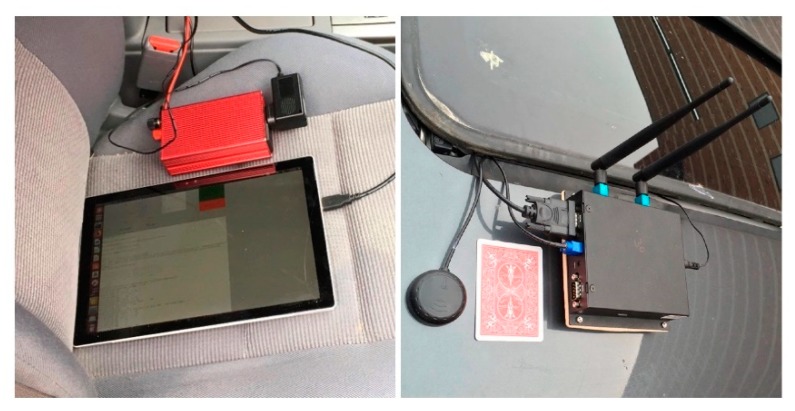
DSRC on-board unit (OBU), GPS antenna, and additional micro-controller (Raspberry Pi) in the DSRC network vehicle experiments.

**Figure 3 sensors-19-03541-f003:**
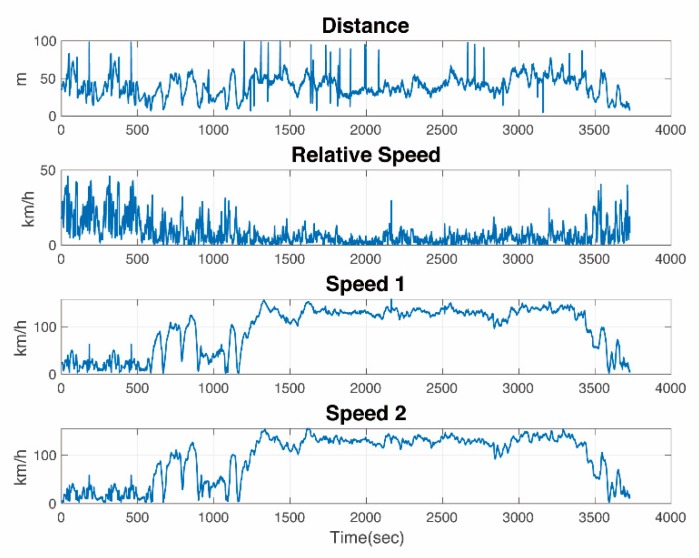
Measurement details (distance and speed) of two vehicles in an experiment in Ann Arbor.

**Figure 4 sensors-19-03541-f004:**
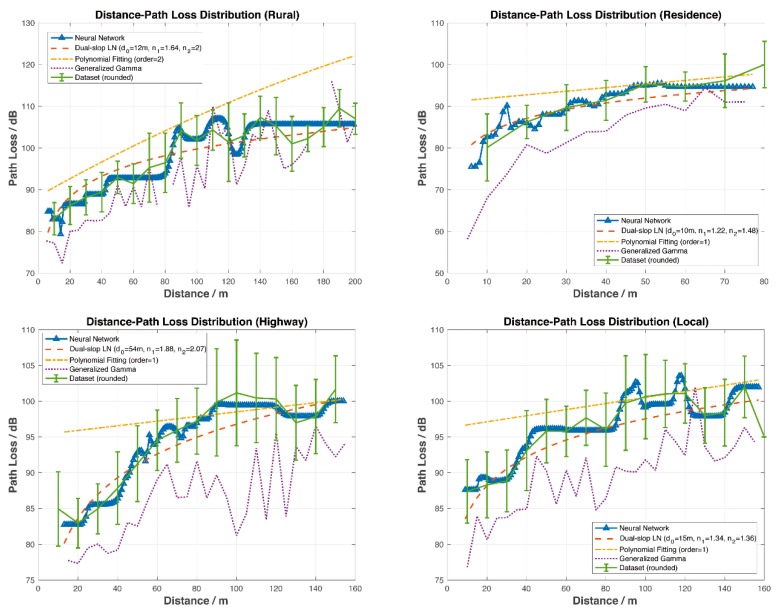
Comparison of four different path loss (PL) prediction models.

**Figure 5 sensors-19-03541-f005:**
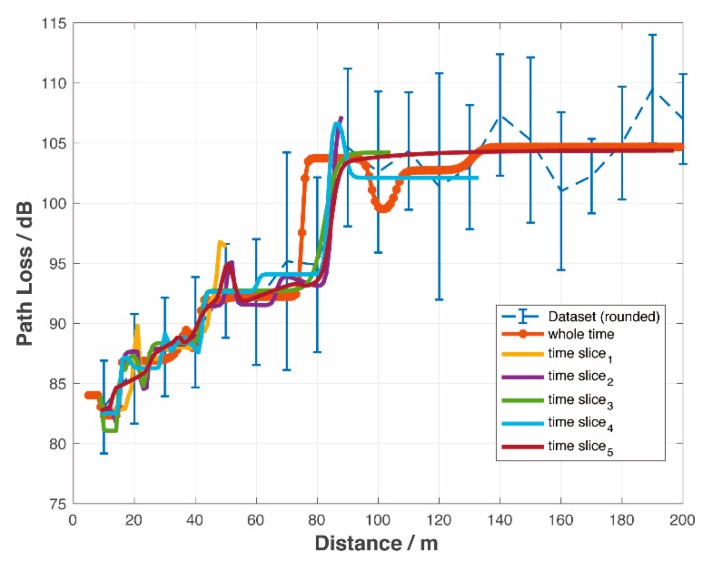
Comparison of the whole-time and real-time PL prediction model based on the NN.

**Figure 6 sensors-19-03541-f006:**
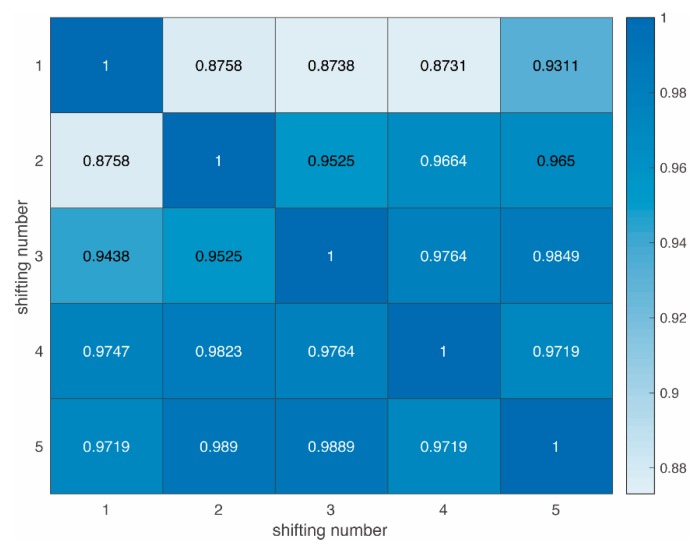
Cross-correlation among different real-time PL predictions.

**Figure 7 sensors-19-03541-f007:**
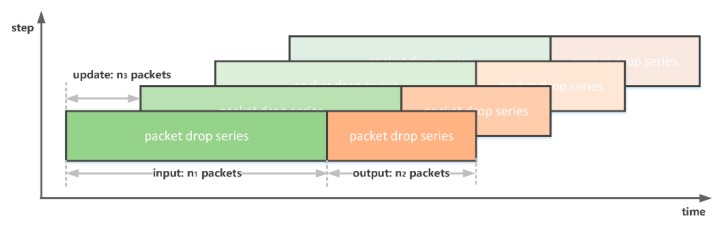
Framework of packet drop rate (PDR) prediction.

**Figure 8 sensors-19-03541-f008:**
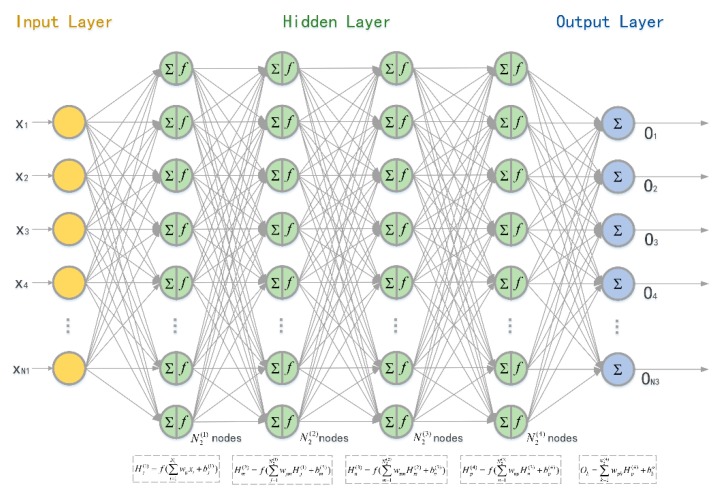
Structure of the neural network for PRD prediction.

**Figure 9 sensors-19-03541-f009:**
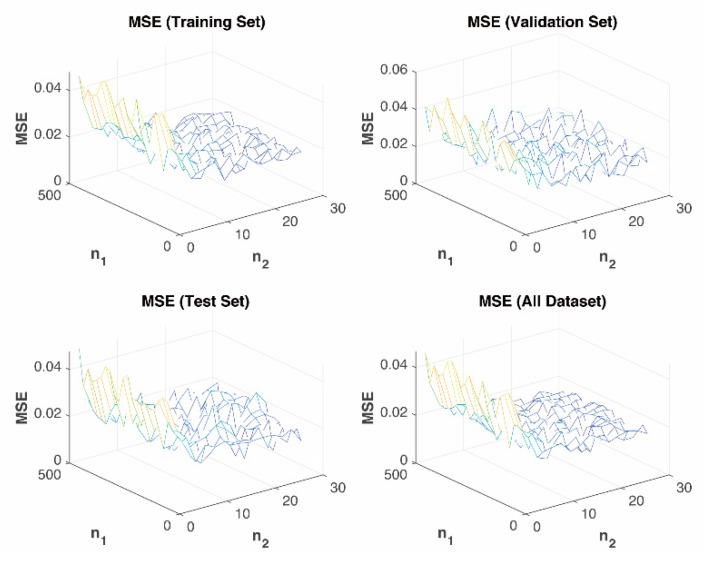
MSE performance of proposed real-Time PD prediction model.

**Figure 10 sensors-19-03541-f010:**
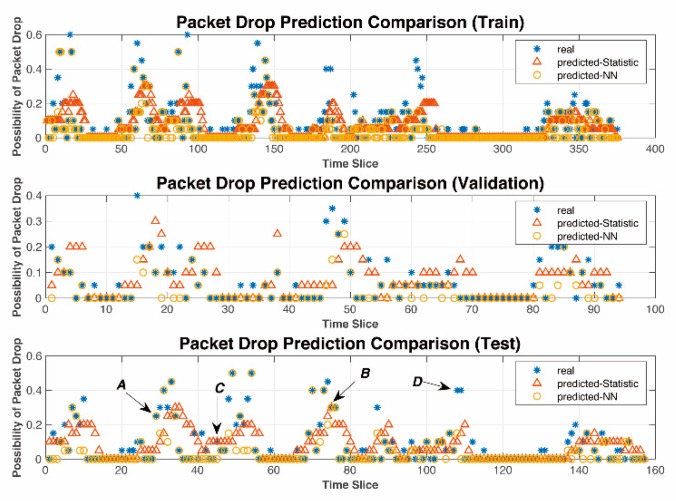
Accuracy performance of proposed real-time PD prediction model.

**Table 1 sensors-19-03541-t001:** Channel fading varies greatly in different regions.

	Maximum (dB)	Minimum (dB)	Median (dB)	Mean (dB)	Standard Deviation (dB)
Highways	112	72	90	90.7923	7.9492
Local Areas	112	72	95	94.5932	6.7879
Residential Areas	110	58	91	90.3498	6.3856
Rural Areas	116	70	85	87.6001	7.8370

**Table 2 sensors-19-03541-t002:** Structure and computations of the neural network.

**Input Data**	**Hidden Layer**	**Output Layer**
$x_{i}$: distance $y_{i}$: path loss	$H_{j}=f(\sum_{i=1}^{N_{1}}w_{ij}x_{i}+b_{j}^{H})$	$O_{k}=\sum_{j=1}^{N_{2}}w_{jk}H_{j}+b_{k}^{O}$
**Squared Error Function**	**Hidden Layer Weights**	**Hidden Layer Weights**
$E=\frac{1}{2}{\left\{\sum_{j=1}^{N_{2}}\omega_{jk}f(\sum_{i=1}^{N_{1}}\omega_{ij}x_{i})-y_{k}\right\}}^{2}$	$\delta w_{ij}=\sum_{k=1}^{N_{3}}(O_{k}-y_{k})w_{jk}(1-O_{k}^{2})x_{i}$	$\delta w_{jk}=(O_{k}-y_{k})H_{j}$

**Table 3 sensors-19-03541-t003:** Statistical characteristics of the dataset used for PL prediction in the residential area.

Distance (m)	Path Loss (dB)			
	Maximum	Minimum	Mean	Standard Deviation
10	97	58	80.1250	8.0274
20	99	74	86.2434	4.0042
30	106	76	89.6890	5.4820
40	104	79	91.4143	4.7768
50	109	84	95.2534	4.2361
60	102	89	94.7170	3.4828
70	110	90	96.1111	6.4118
80	105	92	100.0000	5.5678

**Table 4 sensors-19-03541-t004:** Statistical characteristics of the dataset used for PL prediction in the local area.

Distance (m)	Path Loss (dB)			
	Maximum	Minimum	Mean	Standard Deviation
10	95	72	87.3768	4.4295
20	106	79	88.2857	4.6142
30	102	81	88.8382	4.3174
40	106	80	93.0845	5.5978
50	104	82	95.8140	4.4468
60	102	86	95.7778	3.5261
70	106	85	97.6714	3.8475
80	111	82	96.0159	5.1334
90	111	89	99.7500	6.6114
100	109	89	100.6410	5.8915
110	112	91	101.0256	4.7099
120	109	91	101.0882	4.1514
130	105	91	98.0000	3.8730
140	111	90	98.3947	4.6646
150	109	90	102.0000	4.3028

**Table 5 sensors-19-03541-t005:** Statistical characteristics of dataset used for PL prediction in the rural area.

Distance (m)	Path Loss (dB)			
	Maximum	Minimum	Mean	Standard Deviation
10	96	70	83.0457	3.8613
20	100	76	86.2235	4.5544
30	100	78	88.1616	4.2047
40	104	81	89.4088	4.7390
50	100	84	92.9333	3.8857
60	100	85	91.4545	4.7616
70	108	85	95.2857	8.2606
80	107	84	96.4545	7.1044
90	113	85	104.1562	6.6338
100	111	85	101.8293	5.9745
110	110	92	104.3333	4.8866
120	116	85	101.3846	9.4122
130	111	92	103.0000	5.1547
140	115	97	107.3333	5.0513
150	114	93	105.2500	6.8772
160	108	95	101.0000	6.5574
170	105	98	102.2500	3.0957
180	110	99	105.0000	4.6904
190	116	102	109.5000	4.5056
200	113	104	107.0000	3.7417

**Table 6 sensors-19-03541-t006:** Statistical characteristics of dataset used for PL prediction on the highway.

Distance (m)	Path Loss (dB)			
	Maximum	Minimum	Mean	Standard Deviation
10	96	74	84.9412	5.1898
20	94	72	82.9568	3.4555
30	97	76	84.9444	3.4816
40	98	76	87.8354	5.0656
50	104	80	91.2535	5.3068
60	102	80	94.5312	4.2151
70	106	80	95.9216	4.4670
80	111	82	97.2157	4.6173
90	111	81	99.8387	7.5012
100	109	78	101.1613	7.3986
110	112	80	100.4474	6.2545
120	109	81	100.3143	5.7944
130	105	79	96.9756	5.2321
140	111	82	97.8537	5.1940
150	109	90	101.6757	4.6789

**Table 7 sensors-19-03541-t007:** PDF of the training set (partial).

PDR	Slot 1	Slot 2	Slot 3	Slot 4	Slot 5	Slot 6	Slot 7	Slot 8	Slot 9	Slot 10
0	0.1010	**0.5141**	**0.7770**	0.1493	**0.4073**	0.1006	0.0272	0.1346	0.0031	0.0027
0.05	0.0091	0.1227	0.0306	0.0486	0.1841	**0.4865**	**0.6315**	0.2689	0.0258	0.0689
0.1	**0.6214**	0.1364	0.0469	**0.6910**	0.1469	0.0142	0.1043	0.0094	0.0448	0.0784
0.15	0.0802	0.0544	0.0372	0.0522	0.0814	0.1027	0.0756	**0.2819**	0.0890	0.0539
0.2	0.0066	0.0091	0.0085	0.0049	0.1055	0.1333	0.0397	0.0154	0.0794	0.0280
0.25	0.1133	0.0036	0.0013	0.0135	0.0253	0.0751	0.0161	0.0187	0.0942	0.0258
0.3	0.0335	0.1364	0.0840	0.0089	0.0085	0.0065	0.0232	0.0237	0.0009	0.0022
0.35	0.0076	0.0021	0.0018	0.0045	0.0035	0.0145	0.0044	0.1576	0.0084	0.0102
0.4	0.0043	0.0102	0.0054	0.0010	0.0123	0.0309	0.0243	0.0731	0.0602	0.0682
0.45	0.0008	0.0084	0.0064	0.0013	0.0013	0.0005	0.0077	0.0018	0.0002	0.0013
0.5	0.0159	0.0004	0.0001	0.0034	0.0042	0.0063	0.0113	0.0022	**0.4935**	**0.5798**
0.55	0.0011	0.0004	0.0002	0.0009	0.0121	0.0191	0.0249	0.0020	0.0940	0.0614
0.6	0.0052	0.0019	0.0005	0.0204	0.0077	0.0100	0.0100	0.0107	0.0065	0.0191

**Table 8 sensors-19-03541-t008:** PDF of the validation set (partial).

PDR	Slot 1	Slot 2	Slot 3	Slot 4	Slot 5	Slot 6	Slot 7	Slot 8	Slot 9	Slot 10
0	**0.4128**	0.1069	0.0260	0.0297	**0.9445**	0.0196	**0.6496**	**0.7081**	**0.7426**	**0.6915**
0.05	0.1707	0.1433	0.0606	0.0776	0.0081	**0.6812**	0.2323	0.1752	0.1502	0.1875
0.1	0.0599	0.0090	**0.7560**	**0.8014**	0.0061	0.0258	0.0444	0.0248	0.0285	0.0190
0.15	0.1215	**0.5091**	0.0646	0.0125	0.0221	0.0615	0.0377	0.0593	0.0505	0.0532
0.2	0.0942	0.0042	0.0094	0.0092	0.0031	0.0233	0.0062	0.0051	0.0057	0.0050
0.25	0.0099	0.0031	0.0057	0.0044	0.0036	0.0461	0.0016	0.0011	0.0013	0.0009
0.3	0.0453	0.0406	0.0064	0.0216	0.0096	0.0566	0.0183	0.0150	0.0113	0.0271
0.35	0.0038	0.0183	0.0077	0.0038	0.0012	0.0352	0.0008	0.0014	0.0012	0.0012
0.4	0.0573	0.1572	0.0100	0.0031	0.0008	0.0119	0.0026	0.0038	0.0041	0.0074
0.45	0.0079	0.0040	0.0026	0.0129	0.0004	0.0113	0.0040	0.0041	0.0027	0.0063
0.5	0.0045	0.0011	0.0265	0.0058	0.0000	0.0046	0.0001	0.0001	0.0001	0.0001
0.55	0.0099	0.0013	0.0121	0.0008	0.0001	0.0056	0.0007	0.0007	0.0008	0.0004
0.6	0.0023	0.0019	0.0123	0.0173	0.0005	0.0173	0.0016	0.0013	0.0011	0.0005

**Table 9 sensors-19-03541-t009:** PDF of the test set (partial).

PDR	Slot 1	Slot 2	Slot 3	Slot 4	Slot 5	Slot 6	Slot 7	Slot 8	Slot 9	Slot 10
0	**0.8990**	**0.3699**	**0.6171**	0.0535	0.1934	0.0291	0.1063	0.1026	0.0574	0.0573
0.05	0.0325	0.1297	0.0892	0.1119	**0.4130**	**0.7108**	0.0794	0.0963	**0.4116**	**0.2640**
0.1	0.0207	0.0057	0.0294	**0.6808**	0.0185	0.0994	0.0338	0.0294	0.0202	0.0282
0.15	0.0283	0.3187	0.1835	0.0535	0.1708	0.0192	0.1682	0.0929	0.0987	0.1590
0.2	0.0076	0.0153	0.0198	0.0157	0.0994	0.0257	0.0026	0.0602	0.2247	0.1464
0.25	0.0024	0.0060	0.0088	0.0153	0.0154	0.0016	0.0321	**0.5370**	0.0301	0.0203
0.3	0.0043	0.0506	0.0180	0.0321	0.0113	0.0301	**0.5369**	0.0097	0.0161	0.0086
0.35	0.0017	0.0195	0.0104	0.0033	0.0174	0.0198	0.0111	0.0111	0.0119	0.0581
0.4	0.0011	0.0765	0.0168	0.0103	0.0421	0.0247	0.0144	0.0105	0.0783	0.1462
0.45	0.0006	0.0020	0.0012	0.0090	0.0011	0.0130	0.0089	0.0006	0.0065	0.0019
0.5	0.0001	0.0012	0.0007	0.0068	0.0018	0.0027	0.0015	0.0239	0.0136	0.0525
0.55	0.0005	0.0025	0.0017	0.0028	0.0105	0.0018	0.0011	0.0173	0.0238	0.0316
0.6	0.0013	0.0023	0.0034	0.0051	0.0052	0.0221	0.0037	0.0085	0.0069	0.0259

**Table 10 sensors-19-03541-t010:** Simulation result: RMSE and accuracy of two real-time prediction models.

	NN Model	Statistic Model	Improvement
$n_{1opt}$	140	100	
$n_{2opt}$	20	22	
$n_{3opt}$	10	11	
$RMSE_{train}$	0.0881	0.1050	16.10%
$RMSE_{val}$	0.0722	0.0884	18.33%
$RMSE_{test}$	0.0814	0.1084	24.91%
$RMSE_{all}$	0.0842	0.1036	18.73%
$ACC_{train}$	0.7067	0.3634	94.47%
$ACC_{val}$	0.6915	0.4186	65.19%
$ACC_{test}$	0.6943	0.3916	77.30%
$ACC_{all}$	0.7013	0.3787	85.19%
α	13925	3288

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
