# Peer review of "A Real-Time Channel Prediction Model Based on Neural Networks for Dedicated Short-Range Communications"

_sensors, 2019, doi:10.3390/s19163541_

Round 1

Reviewer 1 Report

Recommended corrections:

Section 2 (and else where in the paper):

        -the word "microcomputer" should be "micro-controller

Section 3: 

        -section 3.1.3 should be removed (Nakagami-m statistics captures             fading envelope and not power).

        -Fig 4: same comment, Nakagami prediction curve should be         removed from the figure (it is clear that it does not match the other models)

        -section 3.2: all the equations in the text should be written as          numbered equations.

        -Description on the proposed NN algorithm is too short. This is unacceptable since this section is the main contribution of the paper. 

        -Authors should include itemized step-by-step descriptions of the simulation of NN/BP algorithm (similar to what was done in section 3.1).

        -MATLAB program or pseudo-code developed should be included

        -Table 2: does not seem to have much value. the information on MSE for the various prediction curves are already shown in Fig 4. It is better to remove that table.

        -New Table: should be added to document NN/BP algorithm computation, starting from input data, then the hidden layer & output layer weights computations, etc.

        -New Table: sample measurement data used by the NN/BP algorithm for the PL prediction shown in Fig 4, presented as "Data set" (dBm) vs. "Distance" (m). This allows verification of the claim in Fig 4.

Section 4: it is hard to verify the claim that the proposed NN algorithm can be applied to predict packet drop rate and latency, when the algorithm itself is not clearly illustrated in the manuscript.

Reviewer 2 Report

The paper is interesting and meaningful for researchers in the same area. However, some changes should be made before publication. The recommendations are the following:

-Line 30: Theoretical models: The sentence “Electromagnetic models are based on Maxwell’s equations while ray-tracing models are based on simulation [27]” should be rewritten, as it is not clear. Ray tracing models are approximations of Maxwell equations and are based on simulations, but they are also electromagnetic models, so it should be rewritten. 

-Line 45-47, there are several works in the literature which uses neural networks for channel prediction, that can be added as references, as for example: “A Ray Launching-Neural Network Approach for Radio Wave Propagation Analysis in Complex Indoor Environments,” IEEE Transactions on Antennas and Propagation, vol. 62, nº 5, pp. 2777-2786, May 2014.

-Line 81, one space is missing in 0.5GHz. 

-Line 82, path loss has been already defined previously with the acronym PL. 

-Line 83, path loss has been already defined previously with the acronym PL. 

-Line 83, the acronym 3GPP was not defined before, so it must be written completely.

-Line 86, path loss has been already defined previously with the acronym PL. 

-Revise also the acronym of Packet drop (PD), because authors define PD in line 49, but they do not use the acronym after that. Revise line 91/ line 94 / line 95/ line 96.

-Line 102-103, revise the last sentence and improve the grammar. 

-Line 106, use NN instead of neural network, as it has already defined. 

-All the acronyms must be revised along the paper, as they are defined previously, but authors do not used them. 

-Line 106, acronym for path loss.

-Line 159, space is missing in 23dBm

-Line 161, space is missing in 10MHz

-Line 208, please add references after the last sentence where you state that your new method has less Root Mean Square Error (RMSE) than conventional models extracted from experimental data.

-Revise Figure 9, the title is “Latency prediction”, not “latency perdition”

Round 2

Reviewer 1 Report

The revision looks reasonable.

There are many works in the literature on path loss prediction using NN. The authors have added one new ref in [3], dated 2014. They should add more recent references from the IEEE xplore, and delete statements in this paper (if any) claiming that NN is used here for path loss prediction for the first time.

Author Response

Response to Reviewer 1 Comments

Point 1: There are many works in the literature on path loss prediction using NN. The authors have added one new ref in [3], dated 2014. They should add more recent references from the IEEE Xplore, and delete statements in this paper (if any) claiming that NN is used here for path loss prediction for the first time.

Response 1: 

Thank you for your advice. We have added 4 references, about PL prediction using NN, into the paper. (See line 100-102 and reference 3, 11, 27, and 35)

And then, we have deleted the statements claiming that NN is used here for path loss prediction for the first time. (See line 136)

Reviewer 2 Report

Authors have addressed my comments correctly. I recommend the paper for publication.

Author Response

Point 1: Authors have addressed my comments correctly. I recommend the paper for publication.

Response 1: 

Thank you very much. We will continue our research in the future.